# From Data to Value: Gaps in Federated Learning Evaluation for Clinical Deployment in Medical Imaging

**Abstract.** Federated Learning (FL) offers a promising solution to the dual challenges of data privacy and multi-institutional collaboration in medical imaging. However, despite strong benchmark performance, FL models rarely reach routine clinical deployment. We hypothesize that this "last-mile" gap stems from a misalignment between current FL evaluation—focused on technical metrics—and the priorities of value-based healthcare (VBHC). We conduct a structured gap analysis comparing current FL practices with VBHC principles and emerging regulatory frameworks. Seven critical deployment axes are identified; six show high-severity gaps, and one a medium-severity gap. Supporting literature is limited: only one axis is backed by strong evidence, three by moderate, one by weak, and two by very weak reviews. Based on these findings and insights from real-world pilots, we propose a practical roadmap to align FL development with clinical and regulatory expectations. By identifying key evidence gaps and outlining actionable next steps, this work aims to inform translational strategies and support the deployment challenges addressed by the BRIDGE Workshop.

**Keywords:** Federated Learning · Value-Based Healthcare · Global regulatory frameworks · Quality Management System · Trustworthy AI

## 1  Introduction

Federated Learning (FL) has emerged as a promising paradigm for collaborative model training without centralising sensitive patient data [1]. In medical imaging, FL is often seen as a privacy-preserving alternative that bypasses legal and logistical barriers to inter-institutional data sharing. Over the past five years, several landmark initiatives have demonstrated the technical feasibility of FL for key medical imaging tasks such as segmentation, classification, and anomaly detection. Notable examples include the FeTS Challenge (2021) for brain tumour segmentation [2], the EXAM study (2021) for COVID-19 prognosis across 20 institutions [3], and a recent peer-reviewed study on melanoma detection using dermoscopic images [4].

Yet, despite its growing popularity, the clinical deployment of FL-based models remains rare. For instance, fewer than 5% of published FL models in medical imaging have been tested in prospective clinical trials or real-world settings [5].

This gap raises important questions about the alignment between FL methods and the broader goals of health systems, particularly those rooted in Value-Based Health Care (VBHC), where clinical relevance, patient-centred outcomes, and cost-effectiveness outweigh technical performance alone [6].

In this framework, technical performance alone is insufficient; models must demonstrate impact on clinical decision-making, quality of care, patient satisfaction, and system sustainability.

This includes cost-utility analyses, longitudinal tracking of clinical benefit, and the routine use of validated patient-reported outcome measures (PROMs) and patient-reported experience measures (PREMs). Instruments such as the EQ-5D-5L and PROMIS-29 quantify health-related quality of life and enable calculation of quality-adjusted life years (QALYs), thereby anchoring economic evaluations in VBHC [7]. PREMs, exemplified by the HCAHPS survey [8] or the NHS Friends and Family Test—capture patients' perceptions of access, communication and overall care experience [9].

For example, *a model may achieve a high AUC in detecting pulmonary nodules, yet fail to improve diagnostic accuracy or workflow efficiency in practice.* Recent work advocates for moving "beyond accuracy" to include explainability, fairness, external validity, and stakeholder engagement as core dimensions of trustworthy AI [10].

Meanwhile, regulatory bodies worldwide, including the United States Food and Drug Administration (FDA) [11], Japan's Pharmaceuticals and Medical Devices Agency (PMDA) [12], the European Union Artificial Intelligence Act (EU AI Act) [13], China's National Medical Products Administration (NMPA) [14], and South Korea's Ministry of Food and Drug Safety (MFDS) [15], are imposing increasingly stringent requirements for real-world validation, continuous safety monitoring, and explainability of AI-driven medical technologies. Nevertheless, many FL studies still prioritise internal technical metrics [16], often overlooking clinical utility, interpretability, equity, and sustainability.

In this preliminary study, we first perform a gap analysis of existing reviews and meta-reviews and, in parallel, distil our own internal lessons learned; together, these insights underpin an initial roadmap to align federated learning with the demands of value-based, regulation-ready healthcare AI.

## 2   Gap Matrix: FL vs VBHC

To identify key misalignments between current FL practice, VBHC expectations and emerging regulatory frameworks, we conducted a structured gap analysis across seven critical axes: regulation/traceability, external validation, clinical co-design, equity/bias, sustainability, evaluation metrics, and clinical integration.

The axes were distilled from recurring topics in the scientific literature, international policy documents (e.g., FDA GMLP, EU AI Act) and real-world deployments reports. For each axis we compared prevailing FL practices with the corresponding VBHC or regulatory requirement and assigned a gap-severity label.

Evidence identification and grading:

1. *Literature search.* For each axis we searched PubMed, Embase, Scopus, Web of Science, and IEEE Xplore (January 2019 – 10 May 2025) using the query ("federated learning" OR "medical imaging AI") combined with axis-specific terms. Eligible records were systematic reviews, scoping reviews, or structured surveys that reported PRISMA-like methods. Searches were run without language filters; however, only English full-text articles were considered for inclusion. One reviewer screened titles, abstracts, and full texts, resolving borderline cases through discussion with the study team.
2. *Selection.* For this preliminary study, we retained the five highest-quality reviews per axis (n = 35).
3. *Gap-severity scoring.* For each axis we tallied the number of eligible primary studies that explicitly highlighted the existence of that gap. High — 3 or more independent studies reported the gap. Medium — one or two studies reported the gap. Low — no published study reported the gap.
4. *Evidence-strength grading* Each review was evaluated with the abbreviated AMSTAR-2 checklist [17]. For every axis we counted the number of High + Moderate reviews: Strong $\geq 3$; Moderate = 2; Weak = 1; Very Weak = 0. Gap severity was then cross-checked against this evidence level to highlight where conclusions rest on limited data. Note that "Medium" or "Low" here refers to the *quality of the supporting reviews*, not to the importance of the gap itself. A gap can be high-severity yet rest on very weak evidence, highlighting an urgent research need.

The results of the analysis are summarised in Table 1. Gap severity was determined according to the predefined rubric; decision logs of all the steps are available in Supplementary A[1].

## 3   Real-World Case Snapshots

Real-world evidence shows both the promise and the pitfalls of FL. On the success side, a multi-centre cardiac-risk study raised every participating hospital's Area Under the Receiver-Operating-Characteristic curve (AUROC) by up to 8% without any data transfer, confirming that FL can lift under-performing sites while preserving privacy [18]. Likewise, the EXAM consortium trained an oxygen-demand predictor for COVID-19 across 20 institutions on six continents, achieved AUROC $> 0.92$ ($\approx$ 16 % better than local models) and validated it prospectively in a regulatory sandbox [3].

Conversely, a recent meta-analysis reported Dice-score drops of up to 25% whenever imaging protocols varied significantly between sites, stressing the still-unresolved non-Independent and Identically Distributed (non-IID) challenge [19]. Even when overall accuracy is retained, fairness gaps can persist: age- and race-dependent error rates were observed in hospital Machine Learning (ML) systems,

---

[1] Anonymized: https://doi.org/10.5281/zenodo.15742397

a risk that vanilla FedAvg does not mitigate unless equity metrics are audited explicitly [20]. These mixed outcomes support our roadmap pillars on heterogeneous validation and fairness-aware auditing.

While the technical soundness of FL algorithms is no longer in doubt, a second wave of large European programmes is translating those algorithms into day-to-day clinical infrastructure. Some notable examples are the following:

*IDERHA* ("Integration of Heterogeneous Data and Evidence towards Regulatory and HTA Acceptance", (2023-28) [21]. The project addresses the data layer by building a discovery platform that is fully aligned with the forthcoming European Health Data Space (EHDS). The EU framework that sets common rules for cross-border sharing and secondary use of health data—and designed to generate evidence acceptable for health-technology assessment (HTA). Another significant effort is *Clinnova*. Clinnova tackles the workflow layer: its cross-border lung-cancer programme combines chest-CT scans with routine clinical data to improve risk assessment, early diagnosis and survival—without moving patient data among several European countries [22]. With the same spirit, the ongoing *EPND* (2021-2026) is building a discovery and access platform for 60+ neuro-degeneration cohorts, with federated analytics, providing the policy scaffolding by funding privacy-preserving infrastructures that harmonise neurodegeneration cohorts and accelerate biomarker discovery and therapeutic trials [23]. Finally, the flagship *EUCAIM* federation delivers the scale-out layer: an ontology-driven common data model and "hyper-ontology" already aligning more than 25 cancer-imaging repositories and enabling validated federated queries in prostate and breast cancer [24].

Overall, these four European programs showcase how data quality, workflow integration, cohort harmonization, and pan-continental federation can transform FL from a technical prototype into a regulated clinical reality. Similar momentum is visible elsewhere: in North America, the *NIH's Bridge 2 AI* initiative is standardizing consent and metadata for multi-modal datasets [32], in Asia-Pacific, Japan's *MED-AI-Cloud* [33] and Australia's *Federated Imaging Network* [34] are building privacy-preserving infrastructures that link tertiary hospitals nationwide. Together, these converging efforts signal a global shift from algorithmic feasibility to real-world, policy-ready deployment of federated medical-imaging AI.

### 3.1   Lessons learned

Experience from several large federated pilots suggests that the technical plumbing (containers, GPUs, secure aggregation) is part of the story; the other part is the strategic design, data work and thoughtful evaluation that makes those tools useful and ethically defensible.

**Data-discovery and metadata depth**. Before any model is trained, sites must expose layered metadata that lets a prospective analyst "zoom" from catalogue-level counts to table- and field-level descriptors (coding system, temporal granularity, missingness, consent tags). Comparing these rich summaries across partners is the fastest way to forecast whether a candidate architecture

will generalise, spotlight hidden selection bias and decide whether extra harmonisation effort is justified. Case studies applying the FAIR framework into their workflows inside hospitals confirm that machine-actionable metadata is what ultimately enables cross-border, privacy-preserving queries in practice [25].

**Common data models: Necessary but not sufficient.** Mapping EHRs to the OMOP-CDM or Digital Imaging and Communications in Medicine (DICOM) remains the most scalable route [27] to interoperable analytics, but the core standard still omits whole modalities and fine-grained units. Imaging projects therefore rely on the new Medical-Imaging CDM tables [26] while community discussions highlight persistent vocabulary gaps (e.g. intraday events, genomics) and the need for site-specific unit agreements before federated code will run reproducibly. In other words, agreeing to "use OMOP" alone does not guarantee semantic alignment. But making the effort of agree on specific ways of each particular federated project, a time consuming essential step.

**Data harmonisation is slow, essential and sometimes impossible** Even with a CDM in place, source data that have degraded through lossy exports, inconsistent units, or missing provenance may never be fully recovered. Budgeting sufficient time and resources for iterative ETL cycles, validation, and re-extraction is therefore critical.

**Regulation, ethics and machine readability.** FL networks increasingly embed consent codes and data-use constraints directly in their metadata using ontologies such as GA4GH's Data-Use Ontology (DUO) [29], allowing access rules to be enforced automatically at query time and audited afterwards—an essential step toward EHDS compliance and trustworthy AI. Projects that ignore this layer risk building technically elegant yet legally unusable solutions.

**Capacity-building and quality culture.** Even the best schema fails without people who can implement it. Initiatives like the EHDEN Academy [28] provide structured learning paths on ETL, SQL and OMOP conventions, certifying both SMEs and data partners to ensure harmonisation work is done once and done right.

Our experience echoes theirs: underestimating data quality and harmonisation tasks is one of the biggest cause of cost overruns in federated roll-outs. Moreover, the lack of standardised, truly interoperable data models means every new site must repeat the same mapping and validation work from scratch, driving up both timelines and costs.

## 4   Discussion

FL has already demonstrated that it can raise accuracy curves and safeguard patient privacy; however, those achievements have not translated into routine care. The explanation lies less in algorithmic maturity than in a persistent misalignment between the questions FL researchers ask and the evidence clinicians, research, data scientist, bioinformaticians, payers and regulators require. Our gap matrix exposes three structural faults.

*First, metric myopia*: internal scores such as Dice or AUROC dominate publications, yet they are silent on cost, time-to-decision, and patient-reported benefit. *Second, siloed design*: prototypes are still built far from the day-to-day realities of radiology worklists and hospital IT, so integration costs surface late and kill momentum. *Third, evidence fragility*: most studies stop at retrospective testing, leaving regulators without prospective or longitudinal data to judge safety. Bridging these faults demands a research programme that treats clinical value and regulatory readiness as primary design constraints, not after thoughts.

Our proposed roadmap converts those constraints into five tangible workstreams. Sequencing these streams in parallel not sequentially may assist on turning FL projects into regulation-ready, value-generating clinical tools rather than polished proofs of concept.

### 4.1   Roadmap for Value-Aligned Federated Learning

To bridge current FL practices and the expectations of VBHC and regulatory frameworks, we propose a roadmap structured around five strategic pillars. These recommendations aim to guide the development of FL systems technically robust, clinically meaningful, ethically grounded, and regulation-ready.

**Engage stakeholders from project inception:** Early and sustained involvement of clinicians, patients, bio-medical-informatics and regulatory experts via co-design workshops, participatory prototyping and iterative feedback loops ensures that FL models address real clinical needs, align with workflow constraints and reflect patient priorities, fostering trust and downstream adoption.

**Expand evaluation beyond technical metrics:** FL studies should go beyond technical scores such as Dice score and AUC, with (A) patient-reported outcomes to capture perceived benefit, such as PROMs and PREMs. (B) health-economic metrics such as cost-utility and time savings; (C) workflow-impact assessments covering clinician burden and decision latency; and (D) decision-curve analysis to gauge clinical utility.

**Validate in diverse and longitudinal settings:** Robust external validation across multiple institutions, patient populations and time periods should include (A) testing in low-resource, high-variability environments; (B) longitudinal performance tracking to detect model drift; and (C) real-world-evidence collection aligned with regulatory expectations.

**Embed equity and sustainability audits:** Evaluate FL systems for impact on health equity and resource use by (A) reporting stratified performance across demographics; include equal-opportunity, demographic-parity, and subgroup-calibration metrics to identify potential disparities in model performance. (B) computing fairness metrics such as equal opportunity and demographic parity; and (C) assessing infrastructure feasibility in under-resourced settings.

**Establish transparent and auditable workflows:** To meet regulatory standards, FL pipelines must be (A) traceable, with version-controlled models, data lineage, audit logs and unit tests; (B) explainable, providing interpretable outputs and prediction rationales; and (C) monitored post-deployment through continuous performance and safety tracking. Such as map each traceability element to ISO 13485 (Quality management systems for medical devices) [30] clauses to strengthen regulatory alignment.

## 5 Conclusions and future directions

FL for medical imaging has moved well beyond proof-of-concept papers to multi-centre pilots, yet a few extra points of AUC no longer convince clinicians or regulators. Future FL systems must prove patient benefit, integrate smoothly into clinical workflows, uphold fairness and satisfy regulatory expectations across the entire product life-cycle.

Our preliminary work with the gap-matrix reveals that, although methodological advances in FL for medical imaging are impressive, the evidential scaffolding that should support clinical deployment is still fragile. Several "high-severity" axes—most conspicuously sustainability reporting and the choice of evaluation metrics—are propped up by very weak or fragmentary literature. This imbalance underscores a dual imperative: (i) to revisit and critically update the existing body of work through rigorous peer-review, and (ii) to commission new, methodologically solid primary studies. In tandem, we call for focused systematic reviews that converge FL, clinical integration, evaluation metrics and environmental impact into an integrated evidence base, ensuring that future guidelines rest on consolidated data rather than isolated proof-of-concept demonstrations.

Important lessons from this early work include: First, early stakeholder co-design pays off: engaging clinicians and patients from day one surfaces workflow and equity issues before they become regulatory blockers. Second, calibrated risk estimates, decision-curve analysis and subgroup reporting reveal true clinical value far better than technical accuracy. Third, sustainability it is becoming an important additional point to consider: transparent $CO_2$-equivalent accounting for each training round is fast becoming a funding and policy requirement [31].

To close those gaps we distilled global guidance: FDA Good Machine-Learning Practice, the EU AI Act and VBHC into a five-pillar roadmap that runs from early co-design through post-market surveillance. The roadmap offers concrete checkpoints (calibration reporting, prospective multi-site monitoring, carbon-ledger disclosure) so developers can reach Software as a Medical Device (SaMD) approval without losing sight of patient-centred value.

This paper is a starting point, not a final map. We aim to convert this into a stronger systematic assessment. We invite experts from additional clinical sites, research teams and policy environments to build a fully grounded, consensus-driven edition of the next iterations of this work. Interested collaborators willing to contribute as independent evaluators, workflow insights or simply critical feedback, are warmly encouraged to contact the authors after the workshop.

**Table 1.** Structured gap analysis of FL practices against the expectations of VBHC and regulatory frameworks.

| Axis | Typical FL Practice | VBHC / Regulatory Requirement | Gap Severity | Evidence Strength | Quick Interpretation |
|---|---|---|---|---|---|
| **Regulation / Traceability** | Heterogeneous reporting; few post-market surveillance or versioning plans. | Auditability, traceability and early regulatory engagement. | High | Strong (4) | Four high/moderate-quality reviews agree— gap and severity are well-substantiated. |
| **External Validation** | Internal validation or similar datasets only; real-world prospective testing scarce. | Multicentric, longitudinal external and real-world validation. | High | Moderate (2) | Two moderate-quality reviews converge; the gap is credible but would benefit from at least one high-quality systematic review. |
| **Clinical Co-design** | Engineering-driven design; minimal structured involvement of clinicians or patients. | Stakeholder engagement from project inception. | High | Moderate (2) | Acceptable evidence base; additional high-quality qualitative or mixed-methods studies would add certainty. |
| **Equity / Bias** | Rarely reports subgroup performance; observed disparities $\geq 5$ percentage points. | Explicit measurement and mitigation of demographic bias. | High | Moderate (2) | Consistent findings across reviews; guidelines for standardised fairness reporting are still lacking. |
| **Sustainability** | Carbon footprint rarely reported; distributed training infrastructure can be costly. | Cost-effective, environmentally responsible solutions. | Medium | Weak (1) | Only one moderate-quality review supports this gap; conclusions should be treated with caution. |
| **Evaluation Metrics** | Relies almost exclusively on Dice/AUC/F1; no PROMs, QALY or economic endpoints reported. | Demonstrate clinical impact and cost-effectiveness (e.g., QALY, PROMs). | High | Very Weak (0) | The gap looks plausible but is underpinned only by low-quality reviews; a rigorous systematic review is urgently needed. |
| **Clinical Integration** | Standalone prototypes with poor FHIR/HL7 interoperability; limited workflow integration. | Seamless 'plug-and-play' integration into HIS/EHR systems. | High | Very Weak (0) | Severity is based on scant, low-quality literature; stronger empirical or review evidence is required. |

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
