# OpenReview forum: "From Data to Value: Gaps in Federated Learning Evaluation for Clinical Deployment in Medical Imaging"
_MICCAI.org/2025/Workshop/BRIDGE — BRIDGE 2025 Poster_

### Official Review · Reviewer_Z368 · 2025-07-22
**Timely work**

**Rating:** 7
**Confidence:** 4

**Review:**

This paper offers a timely and well-executed analysis of the translational gap between federated learning (FL) research and its clinical adoption in medical imaging. Its major strengths include a structured gap matrix across seven axes, a clearly described and systematic review methodology, and an evidence-based scoring framework using AMSTAR-2. Table 1 is well-organized and effectively synthesizes findings. The roadmap and recommendations are supported by real-world experiences, particularly in Section 4.1, which provides valuable insights from major FL initiatives and practical deployment challenges.

The paper could be further strengthened by including more specific implementation examples or measurable success indicators. A schematic diagram illustrating the five-pillar roadmap (perhaps as a flowchart or layered framework) would help communicate the strategic progression from co-design to post-market monitoring more intuitively. But this does not detract from its impact. Overall, the manuscript is thoughtful, methodologically sound, and highly relevant for guiding future work toward regulation-ready, clinically integrated FL systems.

---

### Official Review · Reviewer_kMMZ · 2025-07-23
**Structured Gap Analysis and Roadmap for Clinically Aligned Federated Learning in Medical Imaging**

**Rating:** 6
**Confidence:** 4

**Review:**

This manuscript offers an evaluation of the translational challenges faced by federated learning (FL) in medical imaging, particularly in relation to value-based healthcare and emerging regulatory standards. Its contribution lies in a structured gap analysis that contrasts prevailing FL practices with clinical and policy axes. The writing is clear, precise, and well-organized. The proposed roadmap relies both policy guidance and deployment experience, providing concrete recommendations for aligning FL systems with clinical and regulatory priorities.

Rather than introducing new algorithms, the work reframes the evaluation paradigm for FL by shifting focus from internal performance metrics to external validity and clinical integration. This reframing is original and essential, addressing a significant but underdeveloped space in the literature. The synthesis of regulatory requirements with deployment-informed observations is particularly valuable and moves beyond prior high-level commentaries.

The significance of the work is substantial. As FL moves toward clinical use, its alignment with clinical value and regulatory expectations becomes critical. The proposed roadmap addresses this alignment directly and constructively. The authors are cautious where the evidence base is weak, particularly in evaluation metrics and sustainability, and their transparency reinforces the credibility of the conclusions.

Several key claims, especially in less researched areas like sustainability reporting and integration metrics, rely on limited evidence. While the lack of quantitative synthesis is understandable, it leaves some gaps in the interpretation of prevalence or effect size.

Despite these constraints, the paper presents a good contribution. It provides a shared diagnostic framework, a policy-aligned development roadmap, and a compelling case for reorienting FL research toward clinical and regulatory utility. It is likely to be of broad interest to researchers, clinicians, and regulators aiming to translate federated systems into real-world healthcare impact.

---

### Official Review · Reviewer_Cjfv · 2025-07-25
**Reviewer Comments**

**Rating:** 7
**Confidence:** 4

**Review:**

## Summary
This paper identifies why Federated Learning (FL) models in medical imaging rarely reach clinical deployment despite strong technical performance. The authors conducted a gap analysis comparing current FL practices against Value-Based Healthcare principles and found six high-severity misalignments, particularly around evaluation metrics, clinical integration, and regulatory compliance. They propose a roadmap emphasizing stakeholder engagement, expanded evaluation beyond technical metrics, and transparent workflows to bridge the gap between FL research and clinical deployment.

## Strengths
List 2–3 strong points in the paper (e.g., novelty, clarity, empirical rigor, real-world relevance).
* The paper addresses a critical gap by identifying why technically successful FL models fail to reach clinical deployment.
* The authors provide a rigorous framework for comparing FL practices against VBHC principles and regulatory requirements.
* The work integrates multiple viewpoints including clinical, regulatory, and policy considerations, based on global regulatory authorities to inform their recommendations.

## Limitations or Areas for Improvement
Are there any gaps, unclear sections, missing experiments, or methodological issues?
* The literature search was conducted by a single reviewer without independent verification, potentially introducing selection bias.
* The proposed roadmap is conceptual without empirical testing or quantitative studies to demonstrate its effectiveness.

## Relevance to BRIDGE Workshop Topics
How well does the paper align with the themes of real-world evaluation, robustness, generalizability, or interpretability?
* The paper identifies why FL models fail to transition from research to clinical practice, emphasizing the need for real-world evidence collection that meets regulatory standards.
* While the roadmap mentions explainability requirements, the paper provides less detailed analysis of interpretability challenges, focusing more on evaluation frameworks than technical approaches to model transparency.

---

### Decision · Program_Chairs · 2025-07-25

**Decision:**

Accept (Poster)

**Comment:**

Dear Authors,

Congratulations!

We are pleased to inform you that your paper has been accepted for the BRIDGE Workshop.
Your paper provides a valuable starting point for discussing the implementation and regulatory challenges of federated learning and the gaps that currently exist. Note that your paper was reviewed by three reviewers from different backgrounds: regulatory, research/academic, and industry, and each one provided slightly different perspectives.

Requirements for your final camera-ready submission (due July 30):
* Incorporate reviewer comments and suggestions where appropriate throughout your paper. At minimum, add a discussion section that acknowledges and responds to the key points raised by reviewers
* Ensure your final draft follows standard MICCAI conference and Springer formats and guidelines
* Please submit your camera-ready source file, and any supplementary material you might have.

We look forward to your presentation and the discussions it will generate at the workshop.

Best regards,
Workshop Organizers